# A Prospective Randomized Control Trial of Lingual Frenuloplasty with Myofunctional Therapy in Patients with Maxillofacial Deformity in a Polish Cohort

**DOI:** 10.3390/jcm13185354

**Published:** 2024-09-10

**Authors:** Anna Lichnowska, Adrian Gnatek, Szymon Tyszkiewicz, Marcin Kozakiewicz, Soroush Zaghi

**Affiliations:** 1Private Psychological and Pedagogical Clinic Land of Imagination i, 32 Bardowskiego Str., 95-200 Pabianice, Poland; 2Panaceum Center for Aesthetic Dentistry, 114 Piotrkowska Str., 90-006 Łódź, Poland; 3Department of Maxillofacial Surgery, Medical University of Lodz, 113 Żeromskiego, 90-549 Lodz, Poland; 4The Breathe Institute, 10921 Wilshire Blvd Suite 912, Los Angeles, CA 90024, USA; soroush.zaghi@gmail.com

**Keywords:** ankyloglossia, malocclusion, functional frenuloplasty, mixed-aged groups, interdisciplinary approach, myofunctional therapy, maxillofacial surgery

## Abstract

**Introduction:** There are few publications concerning ankyloglossia in mixed-aged groups utilizing myofunctional therapy and frenuloplasty in patients undergoing orthodontic treatment and maxillofacial surgery. While it is well known that ankyloglossia is mainly diagnosed in babies, research on functional and structural disorders in different age groups is less common. Thus, there is a high need for specific information about the influence and effectiveness of frenuloplasty with myofunctional therapy on the stomatognathic function and final treatment outcome for a wider variety of patients, especially those with maxillofacial deformities. **Aim:** This paper aims to evaluate the impact of lingual frenuloplasty as an adjunct to myofunctional therapy for the treatment of ankyloglossia in children and adults with maxillofacial deformity. **Methods:** Prospective randomized control trial with 155 subjects. Methods were based on visual observation and examination of the oral cavity. There were two groups: myofunctional therapy vs. myofunctional therapy and lingual frenuloplasty. Patients were randomized based on order of entry into the study. χ^2^ test, Kruskal–Wallis, ANOVA, Student’s *t*-test and others were used for statistical analyses. **Results:** The presented protocol with myofunctional therapy and surgical procedures proved to be significantly more effective in improving tongue mobility and stomatognathic functions such as swallowing, breathing, and oral resting postures as compared to the reference group who underwent myofunctional therapy only. **Conclusions:** Lingual frenuloplasty with myofunctional therapy is highly effective in restoring the equilibrium of the orofacial muscles and the skeleton, which is often disturbed and may lead to unstable functional effects among patients considering orthodontic and orthognathic treatments for maxillofacial deformities.

## 1. Introduction

Ankyloglossia, also known as tongue-tie (TT) or tethered oral tissues (TOTs), is a common anatomical defect of the lingual frenulum, which prevents the tongue from having good mobility and range of motion and often affects stomatognathic functions. The term ‘functional ankyloglossia’ is used to characterize limitations of tongue mobility that may or may not be directly attributable to a structural restriction in the lingual frenulum [1]. In the last decade, there has been an increase in the number of tongue-ties diagnosed [2]. However, there is still a lack of well-organized research on adults, especially those presenting with skeletal malocclusion. Moreover, few evidence-based guidelines and consensus statements exist on ankyloglossia in children [3,4,5] that discuss diagnosing and treating the tongue-tie and how it affects the human body. 

Conventional studies have focused more on the anatomy and structure of the lingual frenulum when it comes to assessing tongue-tie rather than the functional mobility and limitations imposed by the restriction. According to N. Mills [6], the structure of the frenulum is compromised of oral mucosa, fascia, and, in some cases, thin fibres of the genioglossus muscle. The floor of the mouth fascia forms a diaphragm-like structure within the arc of the mandible, which leads to its primary role in suspending the tongue, yet also has fascial connections to the rest of the body. As such, it has been demonstrated that the lingual frenulum should be understood more as an extensive entity rather than one distinct structure. The human tongue is a unique muscle hydrostat, and it can change its shape and contour without changing its volume and without any skeletal support. Meanwhile, the shape, contour, and resting position of the tongue within the orofacial complex has influences on the stomagnathic system, and, in turn, impacts the downstream physiology and structural stability of the entire human body.

Some grading scales, such as Kotlow [7], Corryllos [8] and Hazelbaker [9], are found in the published the literature. The Kotlow and the Corryllos scales are based on the appearance of the frenulum, attachment sites and length. Hazelbaker’s scale also examines breastfeeding and neonatal primitive oral reflexes. These grading scales and classifications (which are routinely used by oral surgeons, maxillofacial surgeons, ENTs and dentists) are most commonly applicable to babies and children and generally refer to appearance of the frenulum rather than the biological functions conducted in the oral cavity that may be affected by the lingual restriction. One of the best functional classifications of restrictions imposed by the lingual frenulum is Zaghi’s [10,11] Tongue Range of Motion Ratio (TRMR), which is an example of functionally classifying the degree of lingual restriction by examining the extent of tongue mobility in the vertical dimension within the confines of the oral cavity. The tool is based on a ratio of the vertical extension of the tongue to the incisive papilla (TIP) compared with the maximal interincisal mouth opening. Moreover, the tool uses the tongue range of motion while the tongue is held in suction against the roof of the mouth in lingual palatal suction.

### Global Analysis

The analysis and criteria for evaluating the frenulum in some of the tests and classifications mentioned above often rely on making an assessment on the presence or absence of tongue-tie based primarily on the structure of the lingual frenulum itself. However, a broader global analysis needs to be considered. Several patients with ankyloglossia may present only minor difficulties, while others may compensate for limitations in tongue movement, i.e., lifting the mandible and/or the floor of the mouth [10,11,12]. The function of an individual structure or tissue, of course, can be locally restricted or created, but our tissues or organs usually form a functional and structural whole. Hence, a global approach is worth analysing. This is a strategy that analyses how the same nerves that innervate structures, i.e., the intrinsic muscles of the tongue and the genioglossus muscle, perform their functions, as well as an analysis that includes the tongue as a tissue dependent on proper tension flowing from the fascia, and finally, the tongue as a tissue directly related to the hyoid bone and the mandible, the spatial location of which depends on the correct work of the muscles surrounding the neck, on the proper bone and joint work of the cervical spine or on the correct location and work of the shoulder girdle, of which the scapula, one of the attachments of the infrahyoid muscles, is an inseparable element. For example, the orofacial complex in infants first works as a unit within the whole body, after which differentiation occurs developmentally: head from the body, the jaw from the head, the lips from the jaw, the tongue from the jaw, the tongue tip from the tongue body, the tongue back from the tongue body and the tongue’s lateral margins from the tongue body. All those movements are necessary for higher levels, such as the refined movements required for mature swallowing patterns, biting, oral resting postures, and speech. Therefore, it may be noted that a lack of independent movements in the oral cavity may limit the development of the bony structure, which may directly lead to malocclusion and functional disruption [13]. Research on the Polish population by Dydak et al. [14] unveiled a significant relationship between ankyloglossia and malocclusion or dental abnormalities, constituting 70.3% of the entire study population, where 62% of those with malocclusion or dental abnormalities exhibited a shortened tongue frenulum. In contrast, among individuals without these issues, ankyloglossia was found in only 22%.

The presented approach focuses on diagnoses based on physiological functions of the stomatognathic system, such as swallowing, biting and chewing, as well as the resting postures of the tongue and lips and their impact on the long-term stability of orthodontic and orthognathic treatment. Our approach is unique among many modalities and methodologies as we combine several things to ensure efficacy, safety and interdisciplinarity of lingual functional frenuloplasty with myofunctional therapy in patients suffering from maxillofacial deformities. This paper aims to present a novel Polish approach to diagnosing ankyloglossia in all ages and a pre-and post-treatment protocol with myofunctional therapy in patients with maxillofacial deformity. It also continues the research published in 2021 by Lichnowska and Kozakiewicz [15,16,17].

## 2. Materials and Methods

The presented study was approved by the bioethical committee of the Medical University of Łódź RNN/73/19/KE. The research was conducted according to The Declaration of Helsinki, and the time frame for collecting data was from January 2022 until June 2024. This prospective randomized control study consisted of of 78 consecutive patients in the experimental group and 77 in the reference group. Subjects were randomized based on order of entry into the study. All participants were Caucasian. The treated group (A) consisted of males (*n* = 38) and females (*n* = 40). In the reference group (B), there were 50 females and 27 males. In the treated group, the average age was 18 years old; in the reference group, it was 26 years old. The inclusion and exclusion criteria are presented in Table 1. Patients assigned to groups were randomised based on the day they reported to the clinic.

### 2.1. Statistical Methodology

Statgraphics Centurion version 18.1.12 (StatPoint Technologies, Warrenton, OR, USA) was used for statistical analyses. χ^2^ tests of independence were utilized to find a relationship between two quality variables. ANOVA as normal distribution was found (together with no significant variability in subgroups). The Kruskal–Wallis test was not found to have normal distribution in subgroups (and as the *p*-value was less than 0.05, there was a statistically significant difference amongst the standard deviations in subgroups at the 95% confidence level); the means of pre- and post-op quantitative data were compared by *t*-test (normal distribution), or quantitative data were compared by the Mann–Whitney (Wilcoxon) W-test to compare medians in groups. The *T*-test for paired samples (or sign test if normal distribution was not found) was used for pre- vs. post-treatment data. Detected differences or relationships were assumed to be statistically significant when the *p*-value was less than 0.05. The Kolmogorov–Smirnov test was performed by computing the maximum distance between the cumulative distributions of the two samples to check the normality of the distribution. The minimum number of necessary samples to meet the desired statistical constraints was calculated as 65, with a 95% confidence level and 5% margin error.

### 2.2. Diagnostic Process

The diagnosis process was specially designed for this research by the Orthognathic Speech Therapy Team. Dr. Soroush Zaghi from *The Breathe Institute* was consulted, and the process was conducted by three independent specialists in their medical or paramedical fields, such as speech and language pathologist/orofacial myofunctional therapists, physiotherapists, one oral surgeon and one maxillofacial surgeon. All of the diagnoses were cross-checked and discussed by the cooperating team. The examination was based on an original, unique and specially designed protocol with a myofunctional diagnosis sheet. Every patient was diagnosed in the same order and with the same food pieces and instruments. The first part of the diagnosis was myofunctional; speech and language pathology examination focused on accurately examining patients’ oral cavities, especially the appearance and attachment sites of the lingual, upper lip and buccal frena, according to Zaghi and Baxter [10,11,18]. The next step was to check the presence of dysfunctions of the stomatognathic system, such as wrong resting postures of lips and tongue, the correctness of swallowing, biting and chewing, accompanied by observing breathing type. As those dysfunctions may lay as a foundation for malocclusion, this part of the diagnosis was highly crucial [19]; during the second stage of the diagnosis, patients were asked to subjectively assess the resting postures of their tongue and lips as well as the placement of the tongue during swallowing. Subjective assessment often provided information that showed patients had problems with their proprioception and were unaware of the present dysfunction, i.e., improper tongue resting posture. In the meantime, the speech pathologist, using a cheek and lips retractor, objectively assessed the resting posture of the tongue in delicate nasal breathing to see whether the patient could do it. As a next step, patients were asked to drink 50 mL of still water to examine their swallowing patterns. The swallowing type was marked atypical if patients presented tongue thrust or interdental placement of the tongue. It was also essential to check biting and chewing patterns, so participants were asked to eat a quarter piece of an apple. During this test, it was vital to mark whether the patients were biting with their central incisors or had a compensation pattern, i.e., using their canines or premolars to bite, which could suggest compensations. At this stage, chewing patterns were observed and marked correct or incorrect when unilateral instead of using both sides of the oral cavity. It was also marked whether they chewed with their mouths open, which was also marked incorrect, or closed. After finishing the diagnosis of stomatognathic functions, patients had a quick interview, and on that basis, the pronunciation of Polish phonemes was assessed. Moreover, facial grimaces, neck tension, and other ankyloglossia compensations were marked. Our myofunctional sheet covers additional issues, such as facial symmetry, the general appearance of the tongue on its dorsal surface, the presence of scalloping, tongue overflow, macroglossia, jaw position, extra protrusive movements of the jaw and mentalis strain during swallowing, breathing type, palatal shape and enlarged tonsils [1,19]. All the issues were marked during the myofunctional, speech and language pathology part of the diagnosis.

The second part of the myofunctional diagnosis was a meeting with a physiotherapist who objectively assessed body posture and its defects, such as scoliosis, temporomandibular joints, the symmetry and length of the legs and the position of the pelvis. Muscle and body tension were checked and marked correct or incorrect when any functional or structural deviation was present. For proper therapy tailoring, examining C-spine and temporomandibular movements was important. Those tests included in C-spine were flexion, extension and rotation to both sides; in TMJ, it was opening, protrusion, retrusion and laterotrusion movements both on the right and left side as well as checking if patients are affected by any form of trismus. All performed tests are based on Polish and worldwide validated guidelines in academic textbooks [20,21,22,23].

The third part involved consulting with an oral surgeon and examining the oral cavity, focusing on the lingual, upper lip, buccal frena, and general condition to qualify a patient for the frenuloplasty procedure.

### 2.3. Myofunctional Preparation Protocol before Frenuloplasty

The protocol consists of between four and six visits with the myofunctional therapist. The overall period needed for preparation was eight weeks at minimum. A myofunctional therapist was the same speech and language pathologist for this research, as it is in the Polish scope of practice. Every patient received a printed brochure with instructions on performing oral exercises, stretches and bodywork, a thorough description of the surgical procedure, a possible complication list and an aftercare protocol with post-surgical recommendations.

During the first session, patients were instructed to perform simple tongue exercises three times daily; each task was to be repeated 15 times. The exercises were as follows:Touching the alveolar ridge with the tip of the tongue, close to the incisive papilla, and moving the tongue downwards to the gum ridge without extra jaw movements (to train jaw grading and dissociation).Touching four spots (the cross) on the lower and upper lips with the mouth slightly open: the middle of the upper lip, the corner of the right lip, the middle of the lower lip and the left corner, finishing with lingual palatal suction if it was possible at this time (to enhance jaw dissociation and the patients’ proprioception).Producing clicking tongue sounds (to enhance the ability to perform lingual palatal suction).The cross with the mouth widely open (to stabilise the tongue from jaw dissociation).The brush painting movement: the tip of the tongue moves from the front of the upper alveolar ridge to the back with one plane movement without the jaw closing, coming back and protruding the tongue between open jaws. Then, patients were asked to flex the tip of the tongue and place the tongue on the alveolar ridge (to enhance a more proprioceptive outcome).Lingual palatal suction starts at 15 s and lasts up to 120 s, with the jaws wide open and with delicate nasal breathing (this was extremely important for the surgical procedure itself).

During the first session, patients were also instructed to place their heads in a horizontal plane and stretch their C-spine using flexion, extension, and rotation exercises. Every movement in the C-spine was to be held for 10 s and repeated 10–15 times once a day. The time of the day was not stated. However, most patients reported doing it in the evening.

Patients were instructed to do all the exercises three times daily for eight weeks. During sessions 2–6, the myofunctional therapist controlled the progress and the quality of exercises during every visit and interviewed patients on their adherence and possible problems in performing the exercises. The progress was recorded by taking photos and/or video recordings of the patients doing the exercises. Introducing new exercises was always based on temporary achievements, and their effectiveness was based on tongue range motion ratio, linear measurements of jaw movements and intraoral muscle palpation. In all cases in the treated group from the 3rd session, patients started working on the masseter and temporalis muscles, performing jaw protrusive and lateral movements to gain a better range of motion.

Apart from the oral exercises, patients were instructed to perform intraoral massages. The schedule for massages was based on the timing before the surgery and was as follows: 8 to 4 weeks once a day for 60 to 90 s per side, 4 to 2 weeks once a day from 90 s up to 3 min per side, 2–1 weeks twice a day from 90 s up to 3 min per side.

The first step was to prepare the extraoral area for touch and to relax the masseter and temporalis muscles and the area close to the temporomandibular joints by performing a circular massage with delicate pressure. Secondly, the patient and therapist worked on the submental area, especially the mylohyoid muscle, by pressing and stretching along the jaw and instructing patients to recognise when the muscles were too tight. The next step was to prepare the oral mucosa for touching and relaxing of the superficial floor of the mouth fascia with delicate massaging with the index finger on the floor of the mouth, along the gum ridges. It was crucial, especially in younger patients, to avoid oral aversion. Following this, patients learned to move the tongue, upholding its lateral margins with index fingers up to the palate, then keeping the tongue in lingual palatal suction with delicate massage of the genioglossus muscle without extensive rubbing or touching the frenulum. Finally, patients were asked to perform lingual palatal suction and maintain it for a duration starting from 15 s and progressing up to 120 s. From the fifth session, patients started performing techniques specially designed for stretching the genioglossus muscle and the floor of the mouth facia (FOM), including forklift, manual stretching [24] and hooking techniques. The stretching techniques and repetitions were tailored individually based on noted progress, the subjective feelings of the patient and the grade of ankyloglossia. Only some techniques could be used in all TT grades, as they may have torn the frenulum. In the 6th session, a second myofunctional diagnosis was conducted, and a written description of all dysfunctions, compensations and progress was prepared for the oral surgeon.

### 2.4. The Frenuloplasty Procedure

The procedure was performed on all patients by the same oral surgeon. Frenuloplasty involved using local anaesthesia with 4% Articaine + Adrenaline. The anaesthesia is injected into three spots on the frenulum from the bottom to the top. Consequently, with mosquito forceps, the incision line is marked, and minor vessels are crushed. Then, a horizontal incision in the mucous membrane of the tongue frenulum is made. This is typically performed in the middle of its length or just below it, using scissors and being very careful about salivary glands. If necessary, the superficial and deep layers of fascia are prepared, along with individual fibres of the genioglossus muscle. The freed edges of the mucosa are then usually sutured using absorbable 5.0 sutures. The entire procedure is performed with the tongue placed in lingual palatal suction. The surgeon always operates in loupes with headlight. Intraoperatively, a manoeuvre is performed to stop the floor of the mouth, and the patient is asked to touch the palate with the tip of the tongue to evaluate the range of motion, the tension of the structures of the tongue frenulum, and the need for further preparation. Patients are also asked to perform some free tongue movements and subjectively assess their range of motion and possible changes in the body. The intraoperative assessment is one of the most critical steps in preventing excessive preparation and reducing post-surgical complications.

### 2.5. The Aftercare Protocol

After the surgical procedure, patients are instructed by the operator to rest for a minimum of three days, avoid physical work, alcohol, nicotine, hot and spicy food, speak less and eat smooth foods. They are also administered with non-steroid painkillers such as ibuprofen, dexketoprofen, or paracetamol, accompanied by cold compresses. The painkillers are prescribed to patients as often as needed to eliminate severe pain, along with an anti-swelling medication based on horse chestnut extract for adult patients. On the 3rd day after the procedure, patients meet with the myofunctional therapist to check the healing process and wound contraction. Patients provided feedback on their condition, pain, swelling and tongue movements. All patients are instructed to monitor their condition by taking photos of the tongue and wound site and to send them to the team. During every visit, pictures and recordings are taken for the whole period of the rehabilitation process. The aftercare protocol of active wound management includes the same exercises as before. However, the number of repetitions is tailored individually to the patients based on the healing and the pain feedback. In most cases, patients are asked to perform the exercises three times daily with 10–15 repetitions. For the time when the sutures are on the wound, patients are asked not to produce the clicking sound and not to perform lingual palatal suction intensively, so as not to tear the wound. It was also crucial during the first seven days to tell the patients that they should avoid any other sucking patterns, kissing, as well as licking ice cream, drinking with straws and eating using utensils.

The first entire myofunctional session was conducted 7 to 9 days after the surgical procedure. It was determined that the wound became more contracted as the inflammation period ended. This determination was reached through interviews on subjective feelings during the procedure, pain, mobility, objective range of motion range measurements (ROM) using the TRMR scale, a set of oral exercises and massage with individually tailored techniques comprising manual stretching, forklift, and in some cases, hooking. Patients were also instructed to continue the home-based rehabilitation, including self-massage at least three times a day. To maintain mobility, they were recommended to continue C-spine exercises to enhance the superficial and deep fascia to be more elastic and mobile.

The whole aftercare rehabilitation period was 12 weeks. The myofunctional therapist met with patients every two weeks, measured the ROM, provided massage and active wound care and took photos. During the last session, 12 weeks after the procedure, the final myofunctional examination occurred according to the aforementioned diagnostic procedure.

## 3. Results

The research results focused on the significant differences between the two groups: the treated group (A) with myofunctional therapy and frenuloplasty, and the reference groups without surgical intervention, only with myofunctional therapy (B). In this research, pronunciation improvement was not included in the overall assessment. This study shows the efficacy of frenuloplasty with myofunctional therapy before and after surgical intervention, with additional data concerning malocclusion, stomatognathic dysfunction or improvement in those functions or the bony structure.

Our study shows that the severity of malocclusion impacts the myofunctional assessment and indicates the possible dysfunction in the orofacial region. In this research, significant differences were visible between I and III and II and III Angle’s Class *p*-value = 0.01 (Figure 1). With the more severe malocclusion, patients had more difficulties in stomatognathic functions, such as swallowing or nasal breathing and jaw and tongue resting posture. This may also indicate that even a minor change in the oral cavity may impact some of its functions. Thus, it is essential to note that several patients had difficulty with tongue resting posture before the multidisciplinary approach, not only because of an anatomical defect—the short frenulum—but also because of ineffective coordination and balance on the floor of the mouth muscles (suprahyoid and infrahyoid muscles), masseter, temporalis muscles and lateral pterygoid and their proprioceptive awareness. The data (Figure 2) collected before and after the procedure showed significantly improved tongue resting posture. At the beginning of the treatment, most patients presented a low resting posture. The *p* < 0.001 of the linear trend test justifies the high relevance of the given protocol. In the reference group (B), patients treated only with myofunctional therapy did not show remarkable improvement, but only slightly.

A similar observation concerning lip posture was found (Figure 3). In the treated group (A), the patients presented as lip-separated at the beginning of the treatment. After the treatment, most patients (86%) had normative lip posture, 11% were still separated, and 2% showed another incorrect positioning, often connected with a noxious habit behaviour (*p* < 0.001) or the severity of malocclusion. In the reference group (B), assisted by myofunctional therapy, only 8.5% of patients had improved lip resting posture. 

When analysing the jaw resting posture (Figure 4), it was found that the post-operational group of patients (A) showed a significant improvement in jaw position to the correct one (*p* < 0.001). The reference group (B) had only one patient with good jaw resting posture. Overall, introducing myofunctional therapy and frenuloplasty in the treated group improved the resting postures of the tongue, lips and jaw.

Stomatognathic functions such as swallowing (Figure 5) also showed improvement connected with changes in oral resting posture of the tongue. After the procedure, 90% of the treated group (A) had normative swallowing patterns. Ten percent of the group did not show improvement, mainly due to severe malocclusion, which prevented them from swallowing correctly. The reference group (B) did not improve significantly using myofunctional therapy as the only tool.

Observation showed that, after myofunctional therapy, patients changed their biting pattern (*p* < 0.001), with a strong relationship between myofunctional therapy and new biting patterns (Figure 6). We cannot exclude the hypothesis that the frenuloplasty itself had some impact on biting. Still, based on observation and patient testimonials, it is believed that myofunctional therapy here was the main factor. Almost 72% of patients felt improvement in biting, while only 28% showed no improvement. The lack of progress was also connected with malocclusion severity and the fact that those patients were still waiting for bilateral sagittal split osteotomy (BSSO), Le Fort I (LFI) surgery or double jaw surgery (BIMAX)**.**

The last function that showed significant improvement among the analysed groups was breathing (Figure 7). The changed tongue resting posture facilitated nasal breathing in the treated group. Myofunctional therapy (group B) does not have any spectacular effect compared to the results of the pre-operational and reference groups. This research proved that frenuloplasty is a very effective procedure for establishing proper nasal breathing in patients who previously breathed orally.

The overall assessment (Figure 8 and Figure 9) indicates that myofunctional therapy is effective for pre- and post-frenuloplasty facilitation. It significantly impacts stomatognathic functions if individually tailored and when the team of specialists focuses on quality over quantity. Significant differences exist between the post-operational and pre-operational groups and between the post-operational and reference groups, which justifies the effectiveness of the presented protocol and demonstrates improvement in stomatognathic functions. It also has to be noted that the overall assessment at the beginning of the research showed that functions such as biting, chewing and jaw resting posture are tightly connected to the severity of malocclusion.

## 4. Discussion

This study examines patients with malocclusion and describes the associations between human stomatognathic function disorders, myofunctional therapy, functional frenuloplasty, and maxillofacial deformities. This was another attempt to relate functional abnormalities to other features captured in examining patients (scars, symmetries, aspects of anatomical structure), which are vital as these data concern patients affected by facial skeletal deformities. Apart from contradictory findings within the last 50 years concerning the relationship between malocclusion and myofunctional disorders, orthodontists, maxillofacial surgeons, physiotherapists and speech and language pathologists agree that correct oral behaviours, listed here as the stomatognathic functions, are fundamental in the stability of any treatment in the craniofacial and orofacial regions. As this research focused on ankyloglossia and myofunctional disorders, it is essential to note that over the last 20 years, there has been a high increase in diagnosing this condition, often called overdiagnosis [2], not because more people are born with ankyloglossia but thanks to the development of medicine, myofunctional therapy, speech and language pathology and physiotherapy knowledge. As stated by Merkel-Walsh and Overland [25], treatments for the functional implications of ankyloglossia often focus on oral motor skills that impact feeding, speech and oral resting posture, in addition to bodywork addressing pre- and post-frenuloplasty changes in the sympathetic nervous system and residual muscle strain. Orofacial myofunctional therapy was proven to be an effective way to help patients with tongue-ties [10,11,18]. Moreover, Zaghi [12] states that functional frenuloplasty assists in treating malocclusions, especially maxillary expansion in children. It stabilises the effects of maxillofacial surgeries, which agrees with our current and previous research [15,16,17] as it dramatically affects swallowing patterns and oral resting postures. Lichnowska and Kozakiewicz [15,17] proved that abnormal tongue functioning caused by ankyloglossia affects speech and quality of life in adult patients with skeletal deformities. Moreover, as mentioned above, the authors proved that in most cases, alveolar, dental and palatal phonemes are disturbed. Such a coincidence of malocclusion and speech disorders requires surgical intervention to correct. In both Class II and Class III skeletal defects, similar levels of severity of primary dysfunction were found. The functional frenuloplasty conducted in older children, teenagers and adults assisted in recovering from problems arising from malocclusion. It may prevent other too-short frenulum consequences when the release is made early. Moreover, it improves stomatognathic functions such as mastication, deglutition and chewing. Knosel et al. [26] state that correct resting posture of the tongue is sufficient to prevent, to some degree, obstructive sleep apnoea, as it seems to be more affected by functional than structural conditions. The first person to say that tongue-tie and sleep apnoea coexist was Dr. C. Guilleminault [27,28], who examined the paediatric population and stated that children who underwent functional frenuloplasty with myofunctional therapy had a significant improvement in breathing and fewer incidents of sleep-disordered breathing. In this paper, a mixed-age group was examined, and it was also proved that functional frenuloplasty with myofunctional therapy positively impacts nasal breathing, mainly by changing oral resting postures and enhancing patients’ awareness of their proprioceptive feelings. When considering objective measurements in this paper, the authors deemed that the maximal interincisal mouth opening should agree with standardised ranges of jaw opening movement [20]. If the patient presents a narrow opening, which may suggest jaw trismus, the first specialist to treat the patient should be a stomatognathic physiotherapist. This is crucial for adequately diagnosing tongue-ties and preventing unnecessary surgeries. Our research measurements included a free mouth opening between 37 and 55 (60) mm. Most patients’ records are similar to Zaghi’s [10,11,18,29], suggesting that tongue-tie affects jaw opening, especially when accompanied by malocclusion and temporomandibular disorders. Furthermore, this research proved that a specialist like physiotherapists play a crucial role in preparing a patient for functional frenuloplasty, mainly because the bodywork focuses on the deep frontal fascia layer, which is all over the neck and includes supra and infrahyoid muscles, which control the movements of the tongue. 

According to Sharon Smart et al. and other authors, ref. [30,31] significant heterogeneity in study designs, outcome measures, and care protocols lead to challenges in drawing comparisons across studies. It is, therefore, difficult to determine whether adding pre-operative care may benefit maximising any positive outcomes. However, it appears that pre-operative care does not have any adverse effects. As stated in this research, pre-operative care provided by a team of specialists is not only beneficial for patients. Still, it is also crucial to avoid more severe post-operative complications. However, pre-operative care must be individually tailored and adjusted to patients’ needs, compliance rate and medical condition. The protocol above of pre- and post-care with active wound management prepared by a speech and language pathologist accompanied by a physiotherapist is one of the first in Poland and worldwide in which all necessary details are put together for future implementation anywhere in the world.

There are a few limitations and further ideas to our research. Firstly, the sample group should be more significant in order to understand the outcome better. Secondly, more attention should be paid to the hyoid bone and its impact on possible compensation patterns. Moreover, it would be beneficial to use ultrasound to assess the structure of the tongue before and after the surgical intervention to look for intramuscular scarring and other post-surgical complications which may have adverse effects. The authors also suggest that temporomandibular disorders affect tongue functions, and it will be vital to conduct broader research.

## 5. Conclusions

This research focused on mixed-age patients with tongue-ties who faced severe dysfunctions in the resting postures of the tongue, lips and jaw, abnormal swallowing and breathing discrepancies, and malocclusion. This paper highlighted that pre- and post-care regimens positively enhance functional frenuloplasty and maxillofacial surgery outcomes and do not lead to negative ones. The authors hope that the preliminary research carried out in Poland will prove that speech therapy, myofunctional therapy, stomatognathic physiotherapy and functional frenuloplasty can be of great importance when treating patients with maxillofacial deformities and assist in stabilising outcomes as they target the functional improvement of all stomatognathic functions and structural ones. Despite this, further research is essential.

## Figures and Tables

**Figure 1 jcm-13-05354-f001:**
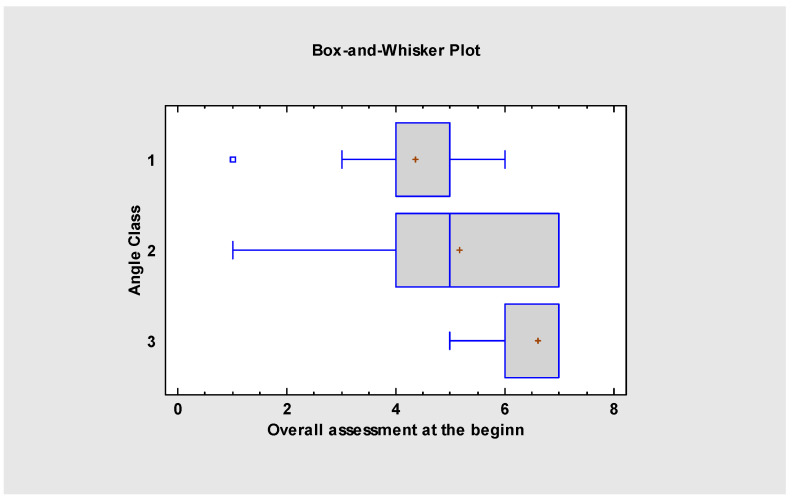
Overall assessment at the beginning by Angle Class.

**Figure 2 jcm-13-05354-f002:**
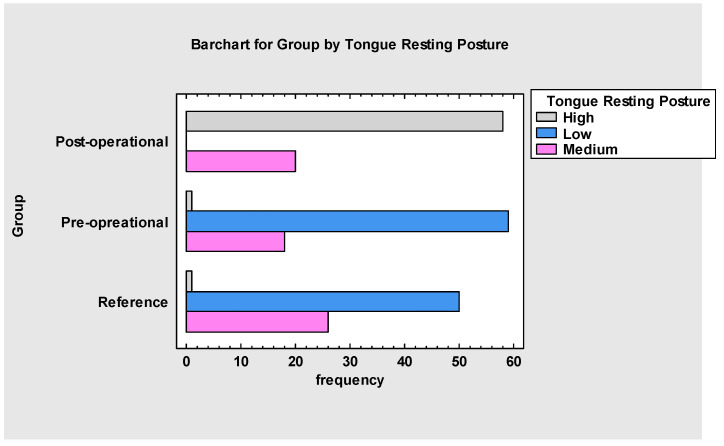
Tongue resting posture before and after.

**Figure 3 jcm-13-05354-f003:**
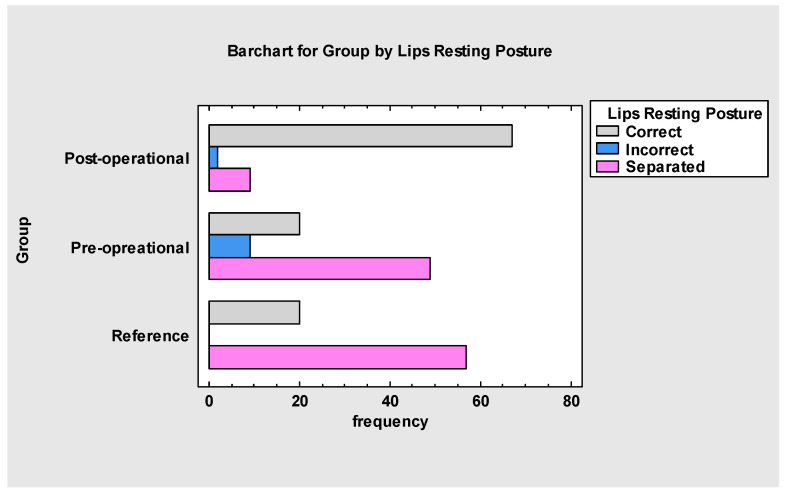
Lips resting posture before and after.

**Figure 4 jcm-13-05354-f004:**
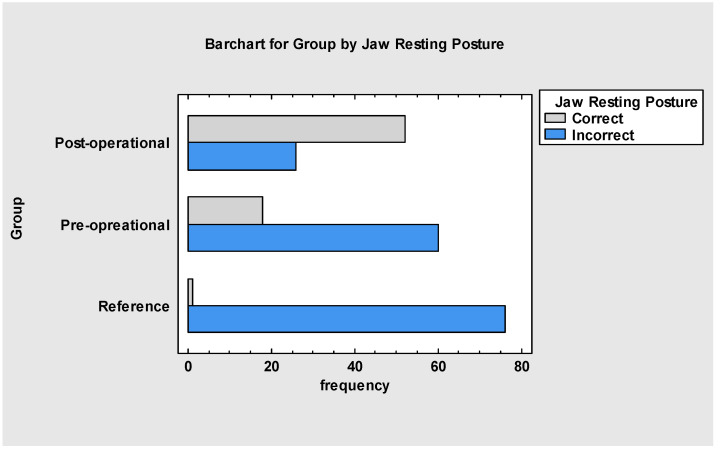
Jaw resting position in all groups.

**Figure 5 jcm-13-05354-f005:**
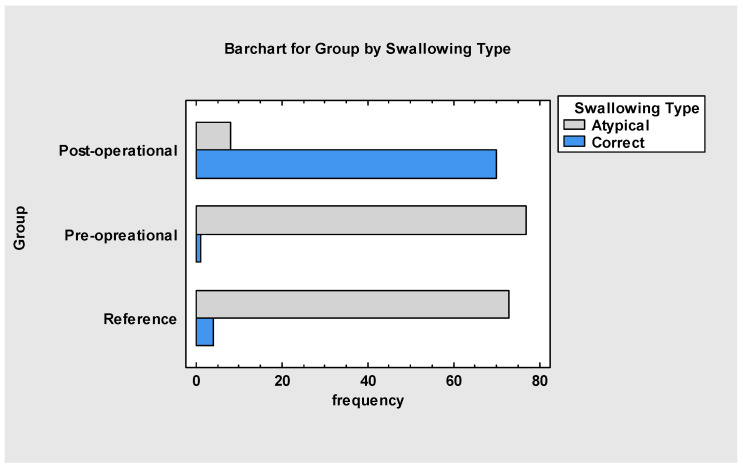
Swallowing type.

**Figure 6 jcm-13-05354-f006:**
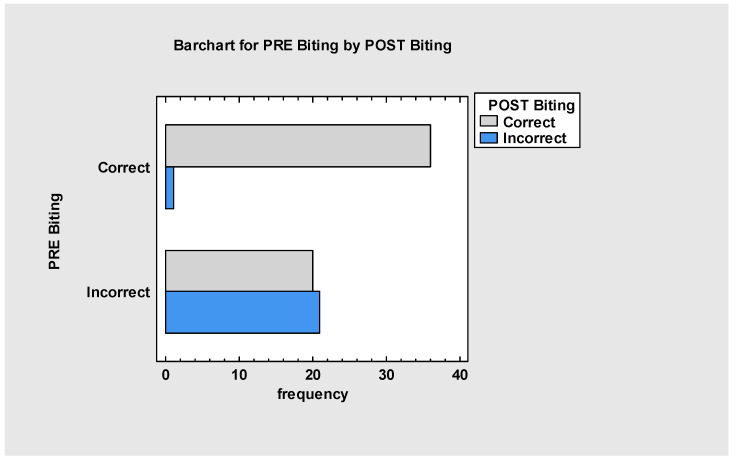
Biting improvement.

**Figure 7 jcm-13-05354-f007:**
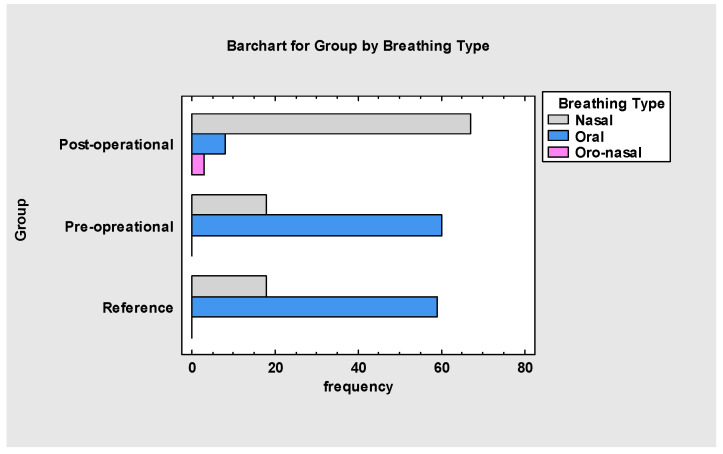
Breathing type in all groups.

**Figure 8 jcm-13-05354-f008:**
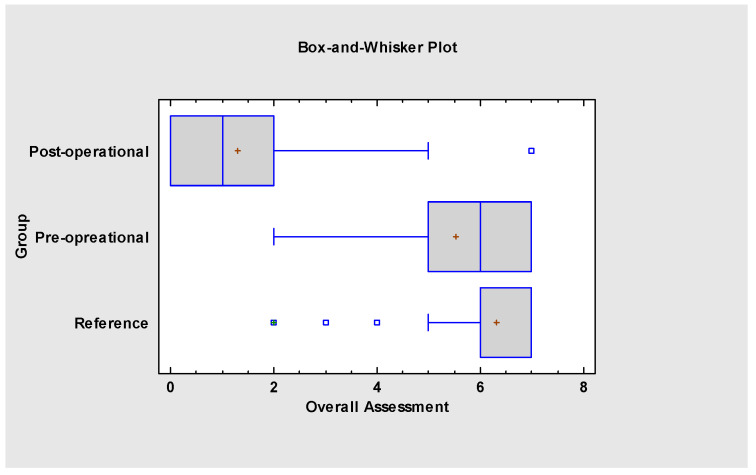
Overall assessment of the project.

**Figure 9 jcm-13-05354-f009:**
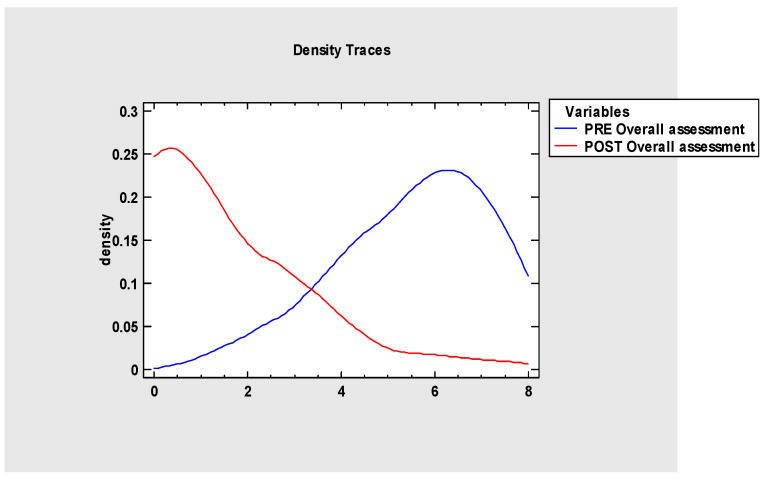
Overall assessment of the project.

**Table 1 jcm-13-05354-t001:** Inclusion and exclusion criteria.

Inclusion Criteria	Exclusion Criteria
Tongue-tied according to TRMR classification	Previous frenotomy or frenuloplasty
Any incorrect stomatognathic function	Previous myofunctional therapy
Class II and III malocclusions	Lack of Class II and III maloclussion
Age 7 to 50 years old	Age below seven years old
Type of surgical treatment: frenuloplasty	Lack of stomatognathic function disruption
Consent for Speech and myofunctional therapy	Surgical treatment of cleft or malocclusion
No previous surgical treatment of the lingual frenulum	Genetic syndromes
Generally healthy	

## Data Availability

The data presented in this study are available on request from the corresponding author due to (specify the reason for the restriction).

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
