# Peer review of "A Prospective Randomized Control Trial of Lingual Frenuloplasty with Myofunctional Therapy in Patients with Maxillofacial Deformity in a Polish Cohort"

_jcm, 2024, doi:10.3390/jcm13185354_

Round 1

Reviewer 1 Report

Comments and Suggestions for Authors

The manuscript titled "The novel approach to frenuloplasty with myofunctional therapy in patients with maxillofacial deformity" has been submitted to the Journal of Clinical Medicine.

The objective of this study was to evaluate the impact of functional frenuloplasty on tongue mobility and functional outcomes in the adult Polish population with maxillofacial deformities. The findings suggest that appropriate pre- and post-operative care regimens can significantly enhance the effectiveness of functional frenuloplasty.

While the manuscript addresses a compelling issue, several concerns regarding the study need to be addressed.

Title: Please describe the type of study. Check grammar. It is recommended to start with "A novel..."

Abstract

- The background should focus on the research problem.

- The objective should not include aspects related to the methodology. There is also no coherence with the title because the novel aspect indicated in it is not described.

- As evidenced in the main text, this is not a population study as indicated in the objective.

- The methodology should present the most relevant aspects related to the stated objectives and their relationship to the novelty described in the title.

- The study groups should be described without including quantities or gender; these should be presented in the results instead.

- The statistical tests are less relevant in the abstract.

- The results should also present concrete values of the variables compared between the groups, including p-values.

- The authors should focus the conclusions based on the results and in coherence with the objectives.

Keywords: Please ensure that all of them correspond to MeSH terms.

Introduction

- The concepts in the first 7 lines must be supported by bibliographic references.

- References 1 and 2 support the same concept and are nearly identical.

- The introduction jumps between several different concepts without a clear, cohesive narrative. It starts with a general description of ankyloglossia, moves to its diagnosis and treatment in infants versus adults, then abruptly discusses anatomical and functional aspects without clear transitions.

- The novelty of the study is not well-articulated. The introduction should clearly state what makes the study unique or innovative compared to existing research.

- The use of terms like "tongue-tie," "tethered oral tissues," and "ankyloglossia" should be consistently defined and used throughout the text to avoid confusion.

- The problem statement lacks depth. The introduction should provide a more detailed background on why ankyloglossia in adults, particularly those with skeletal malocclusions, is an important area of study.

- While anatomical details are important, the introduction spends too much time on the structure of the frenulum and its components without linking this information back to the study’s objectives.

- Claims about the lack of well-organized research and evidence-based guidelines are not adequately supported by references. More citations are needed to substantiate these statements.

- The discussion of different grading scales and methodologies is disjointed and lacks clarity. This section could benefit from a more structured presentation, perhaps with a table summarizing the key features of each scale.

- The introduction includes detailed anatomical descriptions and mentions of specific muscles and their functions, which might be more appropriate for the methods or discussion sections.

- The concept of a "global analysis" is mentioned but not clearly explained. It should be explicitly defined and its relevance to the study should be made clear.

- The objectives of the study are not clearly outlined. It must be consistent with that presented in the abstract.

Methods

- In the first three lines, an objective is described; in addition to being an inappropriate place for this, it is also not consistent with the one described in the introduction and abstract.

- What is described between lines 84 and 88 belongs to the results section.

- Present the type of study.

- The selection criteria for the two groups should be presented.

- The statistical analysis needs to be presented in a more organized manner, starting with descriptive statistics.

- Present the statistical test used to establish the normal distribution of the data along with its corresponding p-value.

- Line 105, include a bibliographic reference.

- It should be clarified if the protocol presented in the subsequent lines was applied to both groups.

- Line 212, clarify if the procedure was performed by the same surgeon.

- Present the sample size calculation.

- The methodology for the diagnostic process is thorough but lacks detail on how the consistency and reliability of the diagnostic assessments were ensured across different specialists.

- The specific protocols and tools used for the myofunctional and speech assessments are described, but there is no mention of validation or standardization for these methods.

- The subjective assessments by patients and the objective evaluations by specialists are not clearly differentiated in terms of how they contribute to the final diagnosis.

- The text provides a detailed description of exercises and their frequency but does not explain how adherence to these protocols was monitored or documented.

- There is no mention of how the effectiveness of the exercises was measured or how adjustments to the protocol were made based on individual patient progress.

-Frenuloplasty Procedure:

· The procedure description lacks information on the criteria for determining the need for additional preparation or adjustments during surgery.

·Details on the qualifications and experience of the surgeons performing the procedures are missing, which could impact the consistency and quality of the surgical outcomes.

-The aftercare instructions are comprehensive but do not address how patient adherence to these instructions was monitored or managed.

- The protocol for pain management and the use of anti-swelling medication are mentioned, but there is no discussion on how the effectiveness of these interventions was evaluated.

- The methodology section could benefit from a clearer organization, with distinct subsections for statistical analysis, sample characteristics, diagnostic processes, intervention protocols, and aftercare.

-There is a lack of discussion on potential limitations of the study design and methodology, which would provide a more balanced view of the research approach (Please include them in the respective section).

Results

-Lines 260-262: In the methodology, the management differences for the two groups are not clear. It is recommended to present this distinction using subtitles to evaluate it more clearly.

- Lines 263-266: Objectives are presented again. Please review. The way the results and the variables being compared are presented was not described in the methodology.

- The section could benefit from clearer organization. While it presents findings, a more systematic layout (e.g., separating results for each function being evaluated) would enhance readability.

- The statistical significance is mentioned (e.g., p < 0.00), but the first whole number should be presented. A value of zero has no meaning.

- The results often compare the treated group with the reference group, but the criteria for these groups should be clearly defined earlier in the methodology to understand the context of these comparisons.

- While figures are referenced, it would be beneficial to provide a brief description of what each figure illustrates. This way, readers can grasp the findings without needing to refer back and forth between the text and figures.

- The interpretation of results sometimes seems implicit rather than explicit. More direct statements about the implications of findings for clinical practice could be included.

- The mention of the severity of malocclusion appears multiple times. It may be beneficial to consolidate these points to avoid redundancy and enhance clarity.

- While conclusions are implied, it may be helpful to explicitly summarize the overall significance of the findings at the end of the results section to reinforce the main takeaways.

Discussion

- The term "functional frenuloplasty" is used multiple times without a clear definition or explanation of its significance. Providing a brief overview of this procedure would enhance understanding for readers unfamiliar with the term.

- The discussion mentions contradictory findings over the past 50 years regarding malocclusion and myofunctional disorders but does not elaborate on these contradictions. A more detailed analysis of these inconsistencies could strengthen the argument and provide a broader context.

- While the discussion touches on the implications for practice, it could be more explicit in connecting the findings to clinical applications. For example, discussing specific treatment protocols or recommendations based on the findings would be beneficial.

- The discussion primarily focuses on clinical outcomes without considering patient experiences or perspectives. Including patient testimonials or qualitative data could provide a more holistic view of the impact of treatments.

- Insufficient Critique of Limitations: Although the authors mention the need for individualized pre-operative care, there is little discussion on the study's limitations, such as sample size or potential biases. Acknowledging these limitations would enhance the credibility of the findings.

- The discussion should suggest potential avenues for future research based on the findings. For instance, exploring long-term outcomes of myofunctional therapy or the impact of different pre-operative care strategies could provide valuable insights.

- The discussion would benefit from clearer organization. Using subheadings to separate different themes or topics would help guide the reader through the section and improve overall readability.

Conclusions

- It should be clarified that the conclusions are presented considering the significant limitations of the study. Additionally, the authors should focus their conclusions based on the results found.

References

- Self-references should be reviewed, as some are unnecessary or out of context.

- It is recommended to use more up-to-date references with the highest level of evidence possible.

- The references do not present the same format.

It is recommended to use a specialized company for language editing and review.

Some sentences are lengthy and complex, which can hinder readability. It is advisable to break them into shorter, clearer sentences to improve comprehension.

Ensure consistent use of terminology throughout the discussion.

The discussion sometimes relies on passive voice, which can make the text less engaging. Where appropriate, consider using active voice to enhance clarity and make the writing more dynamic.

Some phrases could be reworded for greater clarity.

Review punctuation usage, particularly commas. Some sentences may require additional commas to separate clauses or ideas clearly.

Look for phrases that repeat the same idea.

Improve the flow of ideas by using transition words or phrases to connect sentences and paragraphs. This will guide the reader through the discussion more smoothly.

Comments on the Quality of English Language

Extensive editing

Author Response

Dear Reviewer, 

In the atteched docx. file you will find our reply to your comments. 
All changes in the manuscript are marked in green. 

Reviewer 2 Report

Comments and Suggestions for Authors

The authors report on a novel approach to frenuloplasty with myofunctional therapy in patients with maxillofacial deformity. The topic is interesting and relevant to the readers. The study was based on visual observation, oral examination, myofunctional therapy, and surgical procedures. The patients were divided into two groups: one group received only myofunctional therapy, while the other group received both myofunctional therapy and a surgical procedure. The study examined tongue resting posture, lip resting posture, jaw resting posture, swallowing, biting, breathing, and obstructive sleep apnea. They found that pre- and postcare regimens positively influenced the outcomes of functional frenuloplasty and maxillofacial surgery.

Comments: It would have been beneficial to examine the impact of TMI disorders in this study, as that would have made the research more comprehensive. Why was this not done?

Why were patients over the age of 50 excluded?

How was it decided which patients were placed in each study group, beyond meeting the inclusion criteria?

What was the time frame of the study?

I recommend the paper for publication after addressing these questions and making the necessary revisions.

Author Response

Dear Reviewer, 

In the attched docx. file you will find our reply to your comments. 
All changes in the manuscript are marked in green. 

Yours Sincerly, 
Authors

Reviewer 3 Report

Comments and Suggestions for Authors

General Comment: Major Revisions Required

The manuscript presents interesting and relevant findings, but there are several areas where significant improvements are needed to enhance the clarity, accuracy, and completeness of the research. Below is a summary of the required revisions:

Line 2-3: Identify the type of study in the title. The title should reflect whether this is a clinical trial, observational study, review, or another type of research.

Line 21: Correct the notation for the chi-square test; it should be written as "χ²" instead of "X2". Additionally, briefly describe the analysis performed.

Line 23: Rephrase the sentence for clarity. The current phrasing is somewhat unclear and could benefit from simplification or restructuring.

Lines 32-36: Support these statements with relevant references or revise them accordingly to ensure they are backed by scientific literature.

Line 40: Include a reference to oral and maxillofacial surgeons when mentioning oral and medical specialties to ensure comprehensiveness.

Lines 80-83: Specify the type of study conducted in this section to provide context and clarity for the reader.

Line 89: The abbreviation "TRMR" should be expanded upon first use to ensure that all readers understand its meaning.

Lines 95-99: Provide a more detailed explanation of the statistical analysis. Include information on the specific methods used, why they were chosen, and how they contribute to the validity of the findings.

Ethical Approval and Anonymization Process: These sections are entirely missing from the manuscript. Ethical considerations and the process of ensuring participant anonymity must be explicitly stated.

Line 145: Clarify the use of the terms "oral surgeon" and "oral and maxillofacial surgeon". These terms may refer to different specialties in various countries, so it is important to define or distinguish them according to the standards of the country where the study was conducted.

Follow-Up Time and Patient Attrition: Mention the follow-up period for the study and report on the number of patients lost to follow-up, as this is crucial for evaluating the study's validity.

Guidelines for Research Conduct: State the specific guidelines followed in conducting this research. This could include ethical guidelines, reporting standards, or clinical trial regulations

Author Response

(The authors gave the same response as above.)

Round 2

Reviewer 1 Report

Comments and Suggestions for Authors

No additional comments

Comments on the Quality of English Language

minor editing

Reviewer 3 Report

Comments and Suggestions for Authors

All relevant amendments have been performed.